# Implementing the patient partnership approach to quality improvement of care and services: A multiple case study protocol

**Tania Deslauriers**[1,2]*, **Isabelle Gaboury**[3], **Mathieu Jackson**[4], **Brigitte Vachon**[1,2]

**1** School of Rehabilitation, Faculty of Medicine, Université de Montréal, Montreal, Canada, **2** Centre de Recherche de l'Institut Universitaire de Santé Mentale de Montréal, Montreal, Canada, **3** Department of Family Medicine and Emergency Medicine, Faculty of Medicine and Health Sciences, Université de Sherbrooke, Sherbrooke, Canada, **4** Center of Excellence on Partnership with Patients and the Public, Centre de Recherche du Centre Hospitalier de l'Université de Montréal, Montreal, Canada

* tania.deslauriers@umontreal.ca

## Abstract

**Data Availability Statement:** No datasets were generated or analysed during the current study. All relevant data from this study will be made available upon study completion.

### Context

Patient and family partners are being increasingly engaged worldwide in processes aimed at the quality improvement (QI) of healthcare services. There is also growing interest in documenting these engagement processes within organizations to share and improve them. To support the provincial implementation of this approach, the Quebec's ministry of health and social services published, in 2018, the "Framework for the partnership approach between patients, their families and health and social service stakeholders". However, while this framework provides guidelines by describing each partner's role and the ways in which patient and family partners should be engaged in QI processes, it remains unclear how these recommendations were actually used and implemented by different healthcare organizations. The aim of this paper is to present the protocol of a multiple case study that is being conducted to document how this approach was implemented in different large healthcare organizations in Quebec. This study is being conducted in partnership with a patient partner/co-researcher.

### Methods

This qualitative multiple case study will be conducted in four large healthcare organizations in Quebec. Twelve to 15 key respondents will be recruited for each case. Data will be collected from multiple sources: 1) semi-structured individual interviews with the key respondents, 2) non-participant observations of the meetings of the QI committee engaging PFPs and 3) analysis of documents describing the implementation context, vision, structures and/ or processes. The framework method will be used to conduct intracase and intercase qualitative data analysis.

**Funding:** The first author (TD) is supported by a scholarship from the Canadian Institutes of Health Research (CIHR) grant obtained by the directors (BV, IG) (https://cihr-irsc.gc.ca/, no: 159486). She gratefully acknowledges the financial support of the Research Center of the Institut universitaire de santé mentale de Montréal for the writing of this manuscript. The funders did not and will not have a role in study design, data collection, analysis, decision to publish, or preparation of manuscripts.

**Competing interests:** The authors have declared that no competing interests exist.

**Abbreviations:** QI, Quality improvement; CI[U] SSS, integrated [university] health and social services centre (in French, *centre intégré [universitaire] de santé et de services sociaux*); DQEPE, Department of quality, evaluation, performance and ethics (in French, *Direction qualité, évaluation, performance et éthique*); MSSS, Ministry of health and social services (in French, *Ministère de la Santé et des Services Sociaux*); PFP, Patient and family partner (where *family* includes relatives and close friends).

## Discussion

The multiple cases included will allow for comparisons between different ways of engaging PFPs in QI processes within an organization, the factors influencing these practices, their advantages and disadvantages, and their implementation outcomes. The conclusions drawn from this study will allow us to make recommendations regarding PFP engagement in the QI of care and services and to propose implementation examples for other organizations wishing to design and implement PFP engagement initiatives in their context in Quebec or elsewhere.

## Introduction

For more than 20 years now, patient-centered care has been considered an essential component of quality care [1], and patient engagement has become a key priority for achieving this quality [2, 3]. Partnership approaches have emerged in healthcare organizations seeking to move beyond from patient-centered approaches and encourage more active participation from patients, families and communities as partners in the healthcare system [4–6]. The Montreal model characterizes the partnership approach by a dynamic relationship between patients, their families, and health and social services stakeholders, in which each participant's expertise is recognized, shared and pooled in order to co-construct actions that will meet patients' needs while taking into account their life goals [6–9]. Patients and their families are genuine team members in the same way as health and social services stakeholders [5, 6]. Their experiential knowledge related to their experience of illness and use of the healthcare system, their ability to make decisions for themselves with the support of health and social services stakeholders, and their role as caregivers for their well-being are recognized [6]. Their involvement can vary along a continuum from a low level, where patients and their families receive information, to a higher level, where patients and their families are involved as partners [6]. Patient and family partners (PFPs) can become involved at several levels of the healthcare system, particularly in scientific research and in quality improvement (QI) of care and services [6, 10].

### Patient engagement in quality improvement (QI)

QI is a continuous, systematic approach to resolving healthcare problems and improving service delivery to yield better outcomes for patients [11]. As PFPs are increasingly being recognized as stakeholders in the healthcare system, it has thus become essential to partner with them in QI processes [3, 10, 12]. Their experiences with their own health or that of a family member or friend, as well as their care and service trajectories, give them a unique perspective that is both different from and complementary to that of health professionals and managers [5, 12–14]. They are well placed to help identify priority needs and problems encountered in health establishments [15]. They can also help clarify and better define objectives to ensure that they are meaningful for the patients receiving the care and services. They can help develop a change strategy and choose change measurement indicators that are pertinent to patients [16]. PFPs can thus play many roles in planning, designing, advising, questioning, evaluating, recruiting and training, as well as assume leadership functions [17, 18], to improve the quality of care and services.

## Factors that foster or limit PFP engagement in quality improvement

As organizations often have structures in charge of supporting PFP engagement, the implementation of PFP engagement in QI at all hierarchical levels of an organization varies greatly. For example, the roles of each stakeholder and the ways in which PFPs are engaged to improve the quality of care and services are not always clear [16, 19]. The patient partnership approach is still emerging and complex, and needs to be better documented to promote effective partnerships within our organizations [20, 21]. To date, evidence of the effectiveness of patient engagement has mostly concerned patients' engagement in their own individual care [20, 21]. The process and impacts of patient engagement at the organizational levels, such as in QI still need to be documented if we are to improve our understanding of engagement processes that are both realistic and sustainable in different practice settings [16, 21].

Moreover, PFP engagement in QI can be influenced by a number of factors that either foster or limit its implementation [10, 22, 23]. These include strategic support [10, 20, 22, 24], the presence of leaders [16, 23, 25], the presence of formal processes [10, 23, 26] and financial resources [16]. According to several authors, strategic support from senior management, in terms of both vision and organizational culture, is a key factor in effective partnership [10, 20, 22, 24]. They contend that this support promotes the taking into consideration of patients' needs as a key element in decision making, as well as the formalization of mechanisms for engaging and consulting PFPs in the organization's processes. However, organizations have to manage several competing priorities that can hinder patient engagement in QI processes [22, 27, 28], such as implementing electronic medical records [29]. In addition, healthcare programs are generally designed from the perspective of managers and service stakeholders, not that of patients [16]. The partnership approach thus implies having to learn to work differently; it shakes up power relations and professionals' perceptions of the roles that patients and citizens play in the operations and governance of healthcare organizations [20]. Thus, roles and responsibilities within the organization need to be clearly defined. They also need to be backed by a senior management department responsible for supporting the implementation and sustainability of the partnership approach [10, 23, 30] through efforts to foster power sharing and co-leadership at the strategic and organizational levels [20].

The presence of leaders among the healthcare system stakeholders and PFPs at all hierarchical levels also facilitates the adoption, implementation and transformation of engagement practices [16, 23, 25]. It helps establish the necessary infrastructure and promote a change in attitudes and culture to one that is favorable to patient engagement.

Another key element in the implementation of a partnership approach is the development of formal procedures for recruiting, training, and supporting PFPs in QI processes [10, 23, 26]. The PFPs recruited must be diverse and representative of the clientele receiving the care and services [16]. Partnership training and mentoring for all stakeholders ensures a common understanding of the approach and its benefits for the organization [6, 16, 24, 31]. Planning processes are another key factor [23, 32], i.e., adapting practical aspects such as the calendar and time management. Developing a process for evaluating patient engagement and the outcomes is also essential to ensuring continuous improvement of the implementation of the partnership approach [23, 33].

Lastly, having sufficient human and financial resources represents another key factor in enabling both major healthcare innovations and the staff capacity needed to 'co-manage' QI processes with patients [16]. According to Boivin et al., (2018), to advance methods and practices, political leaders must recognize the need for funding, for support and for an infrastructure devoted specifically to the process of engagement [34]. This notion is also supported by

Canfield et al. (2018), who report the need to move beyond simply recognizing PFPs by granting them both formal institutional status and fair and equitable remuneration [17].

## Quebec context

In Quebec, the MSSS made partnering with patients and their families a structural principle of its 2015–2020 strategic plan [35], stipulating that a partnership between patients and the different stakeholders in the healthcare system can help improve the quality of care and services. The Quebec healthcare network is divided into large care and service territories served by 22 integrated [university] health and social services centres (CI[U]SSSs). Each CI[U]SSS must ensure the accessibility, continuity and quality of services intended for the population in its territory, as well as their participation in the management of the health and social services network [35]. To articulate its partnership strategy and support the reorganization of the roles and responsibilities of the different stakeholders in the network's establishments, the MSSS published in 2018 its Framework for the partnership approach between patients, their families and health and social service stakeholders referred to here as the 'Reference Framework' [8]. In most establishments within the network, the department of quality, evaluation, performance and ethics (DQEPE) is the unit designated to implement, ensure the sustainability of and follow up on the partnership approach. However, the Reference Framework also suggests a possible distribution of roles and responsibilities among the different departments and stakeholders in the network establishments (including their partnership offices and their program, nursing, multi-disciplinary services and professional services departments), but leaves this distribution to the discretion of each CI[U]SSS. The partnership offices are responsible for supporting the recruitment and orientation of PFPs who are interested in becoming partners with their CI[U]SSS. These offices are also responsible for training and coaching these individuals, and for constantly managing a 'bank' of PFPs trained to act as partners. For their part, the program departments are responsible for implementing the partnership approach in organizational and clinical practices, including QI. Lastly, the nursing, multidisciplinary services and professional services departments must ensure clinical governance of the partnership approach and its implementation in the QI processes [8].

Thus, the Reference Framework [8] published by the MSSS in 2018 is a useful tool for fostering patient engagement through a partnership approach within organizations in the Quebec healthcare system. However, the reorganization in the field of the roles of each stakeholder in the implementation of this approach and how this approach is implemented can vary from one establishment to the other. As pointed out by Boivin et al. (2018), organizations must not only understand how their environments foster PFP engagement in QI processes, but also how they limit their own capacity to actually do so [34].

## Research objectives

This paper presents the protocol for a study to be conducted in partnership with a patient partner/co-researcher. The aim of this study is to understand how the partnership approach was implemented in different large healthcare organizations in Quebec. It will allow us to achieve these specific objectives for each study case:

1. Describe and compare the vision adopted by each healthcare organization to foster the partnership approach in QI.

2. Describe and compare the implementation of the partnership approach in QI within each healthcare organization.

3. Understand the factors influencing the vision and implementation of the partnership approach in QI within each healthcare organizations.

4. Analyze the adoption, fidelity and penetration of implementation of the partnership approach in QI within each healthcare organization.

## Conceptual frameworks

Two reference frameworks will guide this study: 1) the Reference Framework (Framework for the partnership approach between patients, their families and health and social service stakeholders) [8], and 2) the *Multi-level framework predicting implementation outcomes* [36].

The Reference Framework will be used to analyze the convergences and divergences between the vision and implementation of the partnership approach in QI processes (objectives 1 and 2). It presents the conceptual elements of the partnership approach and the various contexts in which this approach can take shape. It also puts forward conditions for its success and structural actions that can facilitate its implementation. Lastly, it proposes 1) a PFP engagement continuum in the organization of care and services and in governance, 2) a possible distribution of the roles and responsibilities proposed for the different network stakeholders and 3) a logic model illustrating the relationship between the inputs of the partnership approach, the activities and the expected outcomes [8].

The *Multi-level framework predicting implementation outcomes* [36] will be used to categorize the factors that impact PFP engagement in QI processes (objectives 3 and 4). This multi-level framework regards patients as active agents in their own care and as key stakeholders in successful implementation. It is proposed because it includes a specific category for identifying patient-related factors and establishes a link between the different categories of factors and the outcomes of implementation. In our study, it is vital that *patient-level factors* be taken into account, as patients inevitably influence the implementation of their engagement in QI processes. In addition, as already stated by other authors [37, 38], this framework includes four other categories of factors potentially influencing implementation: *innovation*, *provider*, *organizational and structural* [36]. In our study, *innovation* represents the engagement of PFPs in QI processes according to the partnership approach. Thus, *innovation-level* factors encompass those related to the innovation being implemented. *Provider-level* factors concern those related to the individuals implementing the innovation. *Organizational-level* factors include those related to the CI[U]SSS where the *innovation* is being implemented. And, lastly, *structural-level* factors concern those related to the broader sociocultural or community context into which the CI[U]SSS is integrated. We will examine each of these elements in detail, as the five categories of factors will provide a better understanding of all the factors influencing implementation of the partnership approach in QI processes (objective 3). These *causal factors* lead to *implementation outcomes*, which can be evaluated in terms of level of *adoption*, *fidelity*, *implementation cost*, *penetration* and *sustainability* [36]. Only *adoption*, *fidelity* and *penetration* will be analyzed (objective 4) in our study, as the partnership approach is just now emerging in organizations; this would not permit accurate reporting of the *implementation cost* and *sustainability*. *Adoption* signifies the CI[U]SSSs' intention to implement the partnership approach in their QI processes. *Fidelity* signifies the extent to which the partnership approach has been implemented in the QI processes as anticipated by the MSSS. Lastly, *penetration* signifies the perceived integration of the partnership approach into QI processes within the CI[U]SSSs as a whole.

## Methods

### Research design

To better understand the issue under study, we chose a qualitative approach that provides a holistic and in-depth understanding of the phenomenon [39]. The project will take the form of a multiple case study within a constructivist research paradigm [39, 40].

Experts in the field concur that case studies are the best methodology for describing, exploring and understanding a complex phenomenon in its real context, both holistically and from the viewpoints of the persons concerned [40–42]. It therefore lends itself well to the study of management practices [43] designed to promote partnerships with patients and their families. The constructivist research paradigm is also consistent with this phenomenon. Constructivism is based on the notion that there are multiple social realities constructed from participants' individual viewpoints of a given situation [39].

### Definition of a 'case'

In our study, each case represents a large health and social services centre, i.e., a CI[U]SSS, that implemented the partnership approach either prior to or following the MSSS's 2018 publication of its Reference Framework [8]. A CI[U]SSS is part of Quebec's public healthcare network and is associated with one of its administrative regions. It is responsible for ensuring the development and smooth operation of its local health and social services network(s) within its territory. The size of the territory served varies from one CI[U]SSS to another. Those located in large territories must contend with a rural geographic context that can hinder accessibility to care and services. Some health and social services centres, specifically CIUSSSs, have a university affiliation. Along with this affiliation comes a mission of teaching future health practitioners and of developing and applying knowledge generated by scientific research.

### Recruitment and sampling

**The cases.**   A case study should consist of 4 to 10 cases to benefit from the advantages of this design [44]. The cases will be selected by applying a purposive sampling strategy. This strategy is often used to obtain an in-depth understanding of a phenomenon of interest by selecting cases intentionally and strategically based on the study objectives [39]. According to Stake (2005), cases should be selected according to three main criteria: 1) their pertinence to the phenomenon under study, 2) their reflection of a variety of contexts and 3) their provision of good opportunities for learning more about the phenomenon's complexity and contexts [44]. While case pertinence and the possibility of further learning opportunities are recommended criteria in our study, an effort will be made to represent a wide array of CI[U]SSS characteristics, thus providing a better understanding of the phenomenon and the possible variations regarding PFP engagement in QI processes. Four different administrative regions will therefore be targeted. Two cases will have a university affiliation (i.e., two CIUSSSs) and two will not (i.e., two CISSSs). The cases chosen will also vary in terms of the size of territory served.

Next, for each of the CI[U]SSSs identified, the senior management department designated to implement, ensure the sustainability of and follow up on the partnership approach will be contacted by email. A virtual meeting could be held, as needed, to allow the student-researcher to present the research project, answer questions and address concerns voiced by the designated senior management department, and to confirm both the CI[U]SSS's interest in participating in the study and the project's feasibility.

**The participants.**   For each case selected in our study, 12 to 15 key respondents will be recruited. Again, purposive sampling will be used to collect data from individuals who are either informed or experienced with regard to the phenomenon under study [39, 45]. The key respondents are individuals who have contributed to engaging PFPs within their CI[U]SSS in QI, such as those in the DQEPE and partnership offices; planning, programming and research agents; individuals in nursing, multidisciplinary services, professional services and program departments; health professionals; and PFPs. To recruit these participants, a prior meeting will be held with the DQEPE and/or its designated representatives and the research team. The purpose will be to learn about the QI projects engaging PFPs within the CI[U]SSS and to identify the key respondents to be interviewed. Snowball sampling will then be used to identify the other key respondents contributing to PFP engagement in the centre's QI.

## Data collection and procedure

Data will be collected from multiple sources to obtain the perspectives and viewpoints of people who are involved in engaging PFPs in QI processes, as well as a holistic understanding of the phenomenon under study. Three data sources will therefore be used: 1) semi-structured individual interviews with key respondents, 2) non-participant observations of meetings of the QI committee engaging PFPs and 3) analysis of documents describing the implementation context, vision, structures and/or processes. All data will be anonymized and imported into the qualitative data analysis software NVivo 1.7.1. In addition, as suggested by Gale et al. (2013), the student-researcher will keep a log in which she will chronologically record the events in the study, as well as her reflections and impressions throughout the data collection and analysis process [46].

**Semi-structured individual interviews.**   For each case, the key respondents will be interviewed separately or in small groups, depending on their availability and preferences. The participants will also have the option of being interviewed in person, by phone or via web conferencing, again depending on their availability and preferences. The interviews will take an estimated 60 minutes and will be conducted primarily by the student-researcher. Some interviews might be conducted jointly with the patient partner/co-researcher, depending on his availability. Two interview guides have been developed–one for use with healthcare system stakeholders and the other, with PFPs–based on the previously discussed conceptual frameworks [8, 36]. Sample questions are presented in Table 1. The interviews will be audio-taped and transcribed verbatim.

**Non-participant observations during meetings of the QI committee engaging patient and family partners (PFPs).**   For each case, observations made during meetings of the QI committee engaging PFPs will be noted in an observation table. Examples of observable elements are presented in Table 2. The participants in the study will be instructed to continue

**Table 1.  Sample questions from the interview guides.**

| Healthcare system stakeholders | Patient and family partners (PFPs) |
|---|---|
| • What are your roles and responsibilities in the implementation of the partnership approach in QI within the CI[U]SSS?<br>• How do you recruit PFPs to engage in QI within the CI[U]SSS?<br>• How do you train PFPs to engage in QI within the CI[U]SSS?<br>• To what degree and in what way does the CI[U]SSS value and reward implementation of the partnership approach in QI? | ○ What are your roles as a PFP in improving the quality of care and services?<br>○ How were you recruited as a PFP to improve the quality of care and services?<br>○ How were you trained as a PFP to improve the quality of care and services?<br>• How is your engagement as a PFP in improving the quality of care and services valued and rewarded? |

**Table 2. Sample elements observable during meetings of the QI committees engaging PFPs.**

○ Elements related to the planning and running of the meetings (calendar, agenda, speaking time allotted to the PFPs, periods for informal exchanges, etc.).

○ Elements related to the roles and responsibilities of the PFPs and the healthcare system stakeholders (planning, designing, advising, questioning, evaluating, recruiting, training, etc.).

○ Elements related to material resources (minutes of the meetings, information documents, etc.) and to the physical environment (work space, if in-person meetings).

performing their usual work tasks. It is primarily the student-researcher who will make the non-participant observations, but some observations could also be made jointly with the patient partner/co-researcher, depending on his availability.

**Document analysis.** Again for each case, all documents available in the organization in relation to PFP engagement in QI will be collected and analyzed to obtain a better understanding of the implementation context, vision, structures and processes. For example, some organizations may have developed a training guide, a decision support tool or a recruitment procedure. Minutes of the QI committee meetings may also be collected if they provide information on both QI and implementation processes.

## Data analysis

All the data for each case will be analyzed with NVivo 1.7.1 software as a single data corpus, using a combined inductive-deductive approach and based on the seven steps of the Framework method [46]: 1) transcribing the data, 2) becoming familiar with the interviews, 3) coding the data, 4) developing a working analytical framework, 5) applying the analytical framework, 6) charting data into the framework matrix and 7) interpreting the data.

First, the audio-recordings of all the interviews will be transcribed verbatim. Next, the student-researcher will familiarize herself with all the qualitative data collected, review the impressions and reflections she noted in her daily log, and import everything into the analytical software NVivo 1.7.1. She will then read the transcripts attentively, line by line, and begin coding the data for a first case [46]. The first transcripts will be co-coded by two research team members experienced in qualitative analysis and a second student-researcher with research expertise relevant to this study. Next, the codes will be regrouped into categories until a functional, but not definitive, analytical framework has been developed in the form of a tree diagram [46]. This analytical framework will be applied to the subsequent transcripts. Co-coding will continue with the second student-researcher until there are no new emerging themes. In addition, bimonthly meetings will be held with the research team, made up of the two researchers experienced in qualitative analysis and a patient co-researcher. For each meeting, the student-researcher will prepare a presentation summarizing the data analyzed so far, as well as the most revealing verbatim extracts. More ambiguous verbatim extracts can also be presented if the student-researcher feels the need. The need for further data collection may also be discussed. Team members will have the opportunity to express themselves, ask questions and reflect together, in order to reinforce the critical perspective of the student-researcher's interpretations. The data will then be charted into a matrix where it can be managed and summarized [46]. Illustrative citations will be included in the matrix. The last step will consist of interpreting the data [46], analyzing it separately for each case (intracase analysis) and then analyzing it across cases (intercase analysis) [46] to compare the vision and ways in which health organizations engage PFPs in QI processes in different organizational contexts, as well as the factors influencing PFP engagement and the implementation outcomes in actual practice.

## Research in partnership with a patient partner/co-researcher

The engagement of a patient partner/co-researcher in the research team will generate many benefits for this study. First, this person was integrated from the very beginning of the project, allowing him to help determine the research priorities and the pertinence of the study [47, 48]. Next, he contributed to the design of the research protocol. In particular, he has the opportunity to express his views on the research objectives, the key respondents to be recruited and the pertinence of engaging a patient partner/co-researcher in the different stages of the research. His input helped improve the pertinence of the research [49], its methodology and its ethical parameters [47]. Right from the very first meeting, each member of the research team thus expressed their expectations and concerns, and their roles were clarified [50, 51]. While the patient partner/co-researcher will be involved in all stages of the research project, his role may vary from "Listener" to "Decision-maker," depending on subject at hand. The team co-constructed a flexible draft document outlining the contribution expected from him at each stage of the research project [52]. The Involvement Matrix developed by Smits et al. (2020) to support patient and public involvement was used for this purpose [52]. Developed as part of an iterative co-creation process involving PFPs, clinicians and researchers, this tool can be used both prospectively and retrospectively. Table 3 illustrates the contribution expected to date from the patient partner/co-researcher in this study. The horizontal axis shows the rules of engagement in the research project, while the vertical axis shows the stages of the project.

## Rigorous design of the case study

The trustworthiness of this study will be evaluated on the basis of four criteria commonly applied in qualitative research, i.e., credibility, reliability, confirmability and transferability [53].

Its credibility will be strengthened by triangulating the data collection methods (interviews, observations, document analysis), the participants (PFPs, healthcare system stakeholders) and the researchers [53, 54]. The patient partner/co-researcher's engagement in the team meetings will help us to select the most pertinent results and express them in plain language for all patients and healthcare system stakeholders. To support the study's reliability, the use of an audit trail–in the form of detailed notes on each of the contexts–will provide a chronological record of the events in the study, and reveal the methodological and interpretative decisions

**Table 3. Contribution expected from the patient partner/co-researcher in this study.**

| | | Role in research | | | | |
|---|---|---|---|---|---|---|
| | | Listener<br>Is given information | Co-thinker<br>Is asked to give opinion(s) | Advisor<br>Gives (un-) solicited advice | Partner<br>Works as an equal partner | Decision-maker<br>Takes initiative, makes (final) decision |
| **Stage of research** | Determine the research priorities and the pertinence of the study | | | X | | |
| | Determine the research objectives | | | | X | |
| | Determine the research design | X | | | | |
| | Determine the contribution expected from the patient partner/co-researcher in the different stages of the research | | | | | X |
| | Recruit cases and key respondents | | X | | | |
| | Collect data | | | | X | |
| | Analyze the data (coding) | X | | | | |
| | Interpret the data | | | | X | |
| | Write the report | | | X | | |
| | Disseminate the findings | | | | | X |

made throughout the research process [55, 56]. Moreover, the audiotaping and verbatim transcription of the interviews and the use of an analytical software will allow the study to be replicated. The study's confirmability will be strengthened by critical self-reflection achieved through the student-researcher's daily log. In it, she will describe her constructivist position [40], as well as her reflections, impressions, the challenges encountered, and the justifications for any decisions made during the research process. Lastly, to support the study's transferability, detailed and fulsome descriptions of each case and its context will make it possible to determine the transferability of the results to other contexts, based on the study's unique and shared characteristics [40, 53]. Readers will thus be able to make informed decisions about the applicability of the results to their specific context.

## Ethics approval and consent to participate

The study was approved by the Research Ethics Committee of the Centre intégré de santé et de services sociaux de l'Est-de-l'Île-de-Montréal (MP-12-2022-2714). Informed consent will be obtained from all participants during the course of the study. Interview transcripts and meeting observation notes will be de-identified.

Recruitment for this study began on April 26, 2022. The end of the recruitment period is scheduled for April 2024.

## Discussion

### Potential contributions of the project

Conducted in partnership with a patient partner/co-researcher, this project will provide insight into the ways that different large healthcare organizations in Quebec engage PFPs in QI of care and services. The multiple cases included will allow for comparisons between different ways of engaging PFPs in QI processes within an organization, the factors influencing these practices, their advantages and disadvantages, and their implementation outcomes. In other words, this multiple case study will provide a better understanding where, when, how and for whom the engagement of PFPs can be implemented successfully in QI processes, by specifying the necessary and sufficient conditions under which this partnership could achieve the desired outcomes.

The existence of gaps between the actual practices perceived and those hoped for in the MSSS's 2018 Reference Framework [8] will also be highlighted. The CI[U]SSSs involved in this study will benefit from concrete and personalized recommendations, in light of their context, for reducing the gap between current and desired practices. The purpose of these recommendations will be to maximize existing strategies and suggest other possible strategies. Concrete examples of QI initiatives in the field will be documented for each case. These processes will then be compared, highlighting the advantages and disadvantages of each. In this way, organizations will can learn from each other's successes and implement them in their own context if they so wish.

Lastly, recommendations for decision makers will also be made regarding the Reference Framework [8], in order to optimize PFP engagement in QI both in Quebec. These recommendations could also prove useful for other frameworks in other Canadian provinces, for example Ontario's Engagement Patient Framework [57] and elsewhere in the world, for example in Europe: Patient Partnership Framework for the European Reference Networks [58] and in Australia: Consumer, Carer and Community Engagement Framework and Best Practice Guide 2022–2024 [59].

For research, this study will improve knowledge regarding PFPs' engagement in quality improvement, such as better understanding the roles of each stakeholder and how PFPs are

engaged to improve the quality of care and services [16, 19], improving our understanding of organizational-level engagement processes that are both realistic and sustainable in different practice settings [16, 21] and better understanding patients' experience of the engagement process [10].

## Limitations and challenges of the project

The use of the multiple case study as the research methodology for this study entails certain limitations and challenges. First, the researcher must take the time to thoroughly grasp the different epistemological and methodological perspectives of experts in the field of multiple case studies, in order to conduct a rigorous study consistent with his or her paradigmatic stance. The student-researcher must engage in critical self-reflection throughout the study to ensure adherence to the constructivist perspective [40] and transparency about the process.

Another challenge of a multiple case study involving the data analysis stage is ensuring that the data from each data source are analyzed as a single data corpus in order to obtain a better and holistic understanding of the case, rather than treating and reporting each data source individually [60]. The involvement of other members of the research team in this stage will facilitate data integration, thus providing answers to the research questions.

Another challenge to overcome in this study concerns the leeway allowed regarding the diversity of key respondents who will be interviewed and the diversity of the QI processes carried out in each case. Key respondents are the people who contributed to engaging PFPs in their CI[U]SSS in QI. The aim of this study was not data saturation for each stakeholder group, but rather stakeholder diversity in order to obtain complete representation of each case [61]. The preliminary meeting with the DQEPE and/or its designated representatives and the research team for each CI[U]SSS will be essential to clearly identify the first key respondents to be interviewed and to guide the snowball sampling in order to subsequently identify other key respondents.

Lastly, the leeway allowed regarding the diversity of QI processes carried out in each CI[U]SSS represents another element requiring close attention in this study. PFPs can be engaged in a number of roles in the healthcare system [18]. To thoroughly understand PFP engagement in these processes, it is therefore important to adhere to the definition of QI adopted in this study, i.e., a continuous and systematic approach aimed at resolving healthcare problems and improving service delivery to ensure better outcomes for patients [11]. Again, a prior meeting with the DQEPE and/or its designated representatives will be essential in order to thoroughly understand the QI processes engaging PFPs within each CI[U]SSS.

## Conclusion

Partnerships with PFPs have become essential to the QI of care and services. While organizations have structures and tools for fostering patient engagement, the ways in which the roles of each of the stakeholders are reorganized to ensure proper implementation of the partnership approach and the ways in which this approach is implemented in QI processes can vary from one establishment to the other. This multiple case study, conducted in partnership with a patient partner/co-researcher, will provide a better understanding of how healthcare organizations successfully engage PFPs in QI processes in different organizational contexts by helping specify the necessary and sufficient conditions under which this partnership can yield the desired outcomes. The conclusions drawn from this study will allow us to make recommendations regarding PFP engagement in the QI and to propose implementation examples for other organizations wishing to design and implement PFP engagement initiatives in their particular context in Quebec or elsewhere.

## Author Contributions

**Conceptualization:** Tania Deslauriers, Isabelle Gaboury, Mathieu Jackson, Brigitte Vachon.

**Data curation:** Tania Deslauriers, Brigitte Vachon.

**Formal analysis:** Tania Deslauriers.

**Funding acquisition:** Isabelle Gaboury, Brigitte Vachon.

**Investigation:** Tania Deslauriers.

**Methodology:** Tania Deslauriers, Isabelle Gaboury, Mathieu Jackson, Brigitte Vachon.

**Project administration:** Isabelle Gaboury, Brigitte Vachon.

**Software:** Tania Deslauriers.

**Supervision:** Isabelle Gaboury, Brigitte Vachon.

**Validation:** Isabelle Gaboury, Mathieu Jackson, Brigitte Vachon.

**Writing – original draft:** Tania Deslauriers, Isabelle Gaboury, Brigitte Vachon.

**Writing – review & editing:** Isabelle Gaboury, Mathieu Jackson, Brigitte Vachon.

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
