## [Decision Letter · Decision Letter 0]

23 Feb 2024

PONE-D-23-41748Implementing the Patient Partnership Approach to Quality Improvement of Care and Services: A Multiple Case Study ProtocolPLOS ONE

Dear Dr. Deslauriers,

Thank you for submitting your manuscript to PLOS ONE. After careful consideration, we feel that it has merit but does not fully meet PLOS ONE’s publication criteria as it currently stands. Therefore, we invite you to submit a revised version of the manuscript that addresses the points raised during the review process.

We look forward to receiving your revised manuscript.

Kind regards,

Yaodong Gu

Academic Editor

PLOS ONE

Journal Requirements:

"This work is being conducted as part of the first author’s PhD thesis. She gratefully acknowledges the financial support of the Research Center of the Institut universitaire de santé mentale de Montréal for the writing of this manuscript."

"The first author (TD) is supported by a scholarship from the Canadian Institutes of Health Research (CIHR) grant obtained by the directors (BV, IG) (https://cihr-irsc.gc.ca/, no: 159486). The funders did not and will not have a role in study design, data collection, analysis, decision to publish, or preparation of manuscripts."

**Additional Editor Comments:**

The main contribution of this study shall be pointed out in the discussion.

Reviewers' comments:

Reviewer's Responses to Questions

**Comments to the Author**

1. Does the manuscript provide a valid rationale for the proposed study, with clearly identified and justified research questions?

Reviewer #1: Partly

Reviewer #2: Yes

2. Is the protocol technically sound and planned in a manner that will lead to a meaningful outcome and allow testing the stated hypotheses?

Reviewer #1: Partly

Reviewer #2: Yes

3. Is the methodology feasible and described in sufficient detail to allow the work to be replicable?

Reviewer #1: No

Reviewer #2: Yes

4. Have the authors described where all data underlying the findings will be made available when the study is complete?

Reviewer #1: No

Reviewer #2: Yes

5. Is the manuscript presented in an intelligible fashion and written in standard English?

Reviewer #1: Yes

Reviewer #2: Yes

6. Review Comments to the Author

You may also provide optional suggestions and comments to authors that they might find helpful in planning their study.

Reviewer #1: Review comment

This manuscript entitled “Implementing the Patient Partnership Approach to Quality Improvement of Care and Services: A Multiple Case Study Protocol” primarily aimed to present the protocol of a multiple case study that is being conducted to document how this approach was implemented in different large healthcare organizations in Quebec. There are still some problems that cannot up this study to a publishing level. Some suggestions are listed in the specific comments below.

Specific comments:

1. What is the relationship between the multiple case study protocol proposed by the authors and the “Framework for the partnership approach between patients, their families and health and social service stakeholders” published by Quebec’s Ministry of Health and Social Services? Please further emphasise.

2. Given the relevance to the framework, is the current protocol limited to implementation in Quebec? I would recommend that the authors clarify this further in the manuscript (e.g., in the discussion section).

3. “Twelve to 15 key respondents will be recruited for each case”, how was the number of subjects determined?

4. Please provide the full name of the abbreviations when they first appear in the manuscript, e.g., here: “non-participant observations of the meetings of the QI committee engaging PFPs and”. The abbreviation “PFP” is not reasonably defined.

5. “This study will provide a better understanding of how healthcare organizations…”, I would suggest that the authors further enhance the discussion in the abstract section as it is unclear and does not provide information on the significant advantages of the current multiple case study protocol.

6. The research objectives proposed by the authors seem to be overstated, for example, “Analyze the adoption, fidelity and penetration of implementation of the partnership approach in QI within healthcare organizations.”

7. In the methods section, lengthy explanations of some basic research concepts (e.g., “case studies” and “case”) are unnecessary; please simplify them further.

8. In the semi-structured individual interviews, what was the basis for developing the relevant questions in the interview guides?

9. “In addition, since the coding step will be performed mainly by the first student-researcher, bimonthly meetings will be held with the research team…”, please provide more details about the data analysis section, especially the steps involving data coding. How was it implemented?

10. In the rigorous design of the case study, “Its credibility will be strengthened by triangulating the data collection methods (interviews, observations, document analysis) and debriefing the members of the research team during data interpretation via bi-weekly meetings”, How can the credibility of this study be enhanced through “bi-weekly meetings”?

11. In the discussion section, the potential contribution on the project needs to be further strengthened.

Reviewer #2: Review comment

This manuscript entitled “Implementing the Patient Partnership Approach to Quality Improvement of Care and Services: A Multiple Case Study Protocol” primarily aimed to explore how the partnership approach with patients and families is implemented in Quebec's healthcare organizations, following the province's guidelines. The results of this study provide guidance for publilc health. While it is a very interesting topic. But I think this manuscript has a lot of flaws to fill in before it can be published in a journal. There are several questions should be addressed, which list below. I give a minor revision for this manuscript.

Specific comments

1. In the Summary part, "However, while this framework provides guidelines by describing each partner’s role and the ways in which patient and family partners should be engaged in QI processes, it remains unclear how these recommendations were actually used and implemented by different healthcare organizations." Could you elaborate on the specific gaps or uncertainties in the implementation of the guidelines within healthcare organizations that this study aims to explore?

2. "Patient-centered approaches have emerged in healthcare organizations seeking to move away from paternalistic approaches and encourage more active participation from patients, families, and communities as partners in the healthcare system." What are the theoretical frameworks or models that underpin the transition from paternalistic to patient-centered care in the context of QI processes, and how have these models been evaluated for effectiveness in previous studies?

3. "This qualitative multiple case study will be conducted in four large healthcare organizations in Quebec. Twelve to 15 key respondents will be recruited for each case." What criteria were used to select these healthcare organizations and respondents, and how do these criteria ensure a comprehensive understanding of the patient partnership approach across different organizational contexts?

4. "This study will provide a better understanding of how healthcare organizations successfully engage PFPs in QI processes in different organizational contexts..." Based on preliminary insights or existing literature, what are the anticipated challenges in engaging patient and family partners in QI processes, and how might this study's findings contribute to overcoming these challenges?

5. In the Introduction part, “These include strategic support, the presence of leaders, the presence of formal processes and financial resources.” Please add a reference to support this sentence.

6. “It helps establish the necessary infrastructure and promote a change in attitudes and culture to one that is favorable to patient engagement” Please provide a more detailed description of how to address barriers to patient participation, for example, How to Avoid" No Show" to Planned Appointments?

Musa, S., Al Baker, W., Al Muraikhi, H., Nazareno, D., Al Naama, A., & Dergaa, I. (2021). Wellness Program Within Primary Health Care: How to Avoid" No Show" to Planned Appointments? -A Patient-Centred Care Perspective. Physical Activity & Health (2515-2270), 5(1).

7. “For their part, the program departments are responsible for implementing the partnership approach in organizational and clinical practices, including QI.” The reviewer suggested that the author provide a more detailed description of the partnership approach.

8. In the Methods part, “It therefore lends itself well to the study of management practices designed to promote partnerships with patients and their families.” Please add a reference to support this sentence.

9. In the Conclusion part, Please show detailed findings about this manuscript as well as what are the contributions for future clinical or scientific research.

7. PLOS authors have the option to publish the peer review history of their article (what does this mean?). If published, this will include your full peer review and any attached files.

Reviewer #1: No

Reviewer #2: **Yes: **Zixiang Gao

---

## [Author Response · Author response to Decision Letter 0]

8 Apr 2024

Response to Reviewers

Journal Requirements:

The entire manuscript has been revised to follow the manuscript body formatting guidelines.

"This work is being conducted as part of the first author’s PhD thesis. She gratefully acknowledges the financial support of the Research Center of the Institut universitaire de santé mentale de Montréal for the writing of this manuscript."

"The first author (TD) is supported by a scholarship from the Canadian Institutes of Health Research (CIHR) grant obtained by the directors (BV, IG) (https://cihr-irsc.gc.ca/, no: 159486). The funders did not and will not have a role in study design, data collection, analysis, decision to publish, or preparation of manuscripts."

My amended statements have been included in my cover letter.

My amended statements have been included in my cover letter.

My ethics statement appears at the end of the "Methods" section of my manuscript and does not appear in any other section.

Additional Editor Comments:

The main contribution of this study shall be pointed out in the discussion.

Reviewers' comments:

Reviewer's Responses to Questions

Reviewer #1: Review comment

This manuscript entitled “Implementing the Patient Partnership Approach to Quality Improvement of Care and Services: A Multiple Case Study Protocol” primarily aimed to present the protocol of a multiple case study that is being conducted to document how this approach was implemented in different large healthcare organizations in Quebec. There are still some problems that cannot up this study to a publishing level. Some suggestions are listed in the specific comments below.

Specific comments:

1. What is the relationship between the multiple case study protocol proposed by the authors and the “Framework for the partnership approach between patients, their families and health and social service stakeholders” published by Quebec’s Ministry of Health and Social Services? Please further emphasise.

The Reference Framework (Cadre de référence de l’approche de partenariat entre les usagers, leurs proches et les acteurs du système de santé et de services sociaux) is one of two reference frameworks guiding the conduct of this study, more specifically for objectives 1 and 2. This Framework was published by the Quebec Health and Social Services Ministry in 2018 to support the implementation of the partnership approach within the organizations (CI[U]SSS) of the health and social services network.

The conclusions drawn from this study will allow us, among other things, to formulate recommendations to the Ministry on how the framework could be improved and how organizations can be supported to improve engagement of PFPs in QI.

2. Given the relevance to the framework, is the current protocol limited to implementation in Quebec? I would recommend that the authors clarify this further in the manuscript (e.g., in the discussion section).

Our study has several other potential contributions that will not be limited to the Quebec context.

As mentioned in the Rigorous design of the case study section, detailed and fulsome descriptions of each case and its context will make it possible to determine the transferability of the results to other contexts. Readers will thus be able to make informed decisions about the applicability of the results to their specific context.

There are other reference frameworks in other Canadian provinces, e.g. Ontario’s Engagement Patient Framework (https://www.hqontario.ca/Portals/0/documents/pe/ontario-patient-engagement-framework-en.pdf) and elsewhere in the world, e.g. in Europe: Patient Partnership Framework for the European Reference Networks (https://download2.eurordis.org/ern/Patient-Partnership-Framework/2023/PPF.pdf) and in Australia: Consumer, Carer and Community Engagement Framework and Best Practice Guide 2022-2024 (https://www.seslhd.health.nsw.gov.au/sites/default/files/groups/Planning_Population_and_Equity/docs/Consumer_SESLHD_Digital.pdf). These reference frameworks have points of similarity that will make the results transferable.

This point has been clarified in the manuscript, in the discussion section, as suggested.

3. “Twelve to 15 key respondents will be recruited for each case”, how was the number of subjects determined?

Twelve to 15 respondents was considered an average target number of interviews sufficient to gather relevant information for each case (Malterud, Siersma & Guassora, 2016). However, it is difficult to confirm the exact number of interviews required. It is important for the research team to remain flexible to decrease or increase the number of interviews to document in sufficient depth the implementation used for each case.

4. Please provide the full name of the abbreviations when they first appear in the manuscript, e.g., here: “non-participant observations of the meetings of the QI committee engaging PFPs and”. The abbreviation “PFP” is not reasonably defined.

The abbreviation PFPs stands for patient and family partners. It was first defined in the introduction section. It is also defined in Table 1. It has been added to the title Non-participant observations during meetings of the QI committee engaging patient and family partners (PFPs) as suggested.

5. “This study will provide a better understanding of how healthcare organizations…”, I would suggest that the authors further enhance the discussion in the abstract section as it is unclear and does not provide information on the significant advantages of the current multiple case study protocol.

The discussion in the summary section has been improved. 

See lines 50 – 55: The multiple cases included will allow for comparisons between different ways of engaging PFPs in QI processes within an organization, the factors influencing these practices, their advantages and disadvantages, and their implementation outcomes.

6. The research objectives proposed by the authors seem to be overstated, for example, “Analyze the adoption, fidelity and penetration of implementation of the partnership approach in QI within healthcare organizations.”

It was clarified that research objective will be achieved only for the four cases included in the study.

7. In the methods section, lengthy explanations of some basic research concepts (e.g., “case studies” and “case”) are unnecessary; please simplify them further.

In the methods section, some basic research concepts (e.g., “case studies”, “constructivist research paradigm” and “case”) have been simplified.

8. In the semi-structured individual interviews, what was the basis for developing the relevant questions in the interview guides?

Two interview guides have been developed – one for use with healthcare system stakeholders and the other, with PFPs – based on two conceptual frameworks (the Reference Framework (Cadre de référence de l’approche de partenariat entre les usagers, leurs proches et les acteurs du système de santé et de services sociaux) and the Multi-level framework predicting implementation outcomes). The questions in both guides are similar. The questions in the PFP guide have been simplified with the help of the patient co-researcher. The questions are divided to address all four objectives, i.e. vision (e.g. definition of the partnership approach, priority given by the organization, etc.), implementation (recruitment process, training, etc.), influencing factors (5 categories: innovation, patient, provider, organizational and structural) and outcomes (adoption, fidelity and penetration).

9. “In addition, since the coding step will be performed mainly by the first student-researcher, bimonthly meetings will be held with the research team…”, please provide more details about the data analysis section, especially the steps involving data coding. How was it implemented?

More details have been added to the data analysis section, especially the steps involving data coding, as suggested.

10. In the rigorous design of the case study, “Its credibility will be strengthened by triangulating the data collection methods (interviews, observations, document analysis) and debriefing the members of the research team during data interpretation via bi-weekly meetings”, How can the credibility of this study be enhanced through “bi-weekly meetings”?

The sentence on triangulation has been modified to clarify this justification. Debriefing has been removed from the sentence.

11. In the discussion section, the potential contribution on the project needs to be further strengthened.

The potential contribution to the project has been strengthened. See the discussion section.

Reviewer #2: Review comment

This manuscript entitled “Implementing the Patient Partnership Approach to Quality Improvement of Care and Services: A Multiple Case Study Protocol” primarily aimed to explore how the partnership approach with patients and families is implemented in Quebec's healthcare organizations, following the province's guidelines. The results of this study provide guidance for publilc health. While it is a very interesting topic. But I think this manuscript has a lot of flaws to fill in before it can be published in a journal. There are several questions should be addressed, which list below. I give a minor revision for this manuscript.

Specific comments:

1. In the Summary part, "However, while this framework provides guidelines by describing each partner’s role and the ways in which patient and family partners should be engaged in QI processes, it remains unclear how these recommendations were actually used and implemented by different healthcare organizations." Could you elaborate on the specific gaps or uncertainties in the implementation of the guidelines within healthcare organizations that this study aims to explore?

The Reference Framework suggests a possible distribution of roles and responsibilities among the different departments and stakeholders in the network establishments. For example, the partnership offices are responsible for supporting the recruitment and orientation of PFPs who are interested in becoming partners with their CI[U]SSS. These offices are also responsible for training and coaching these individuals, and for constantly managing a ‘bank’ of PFPs trained to act as partners. But how PFPs are recruited, trained, etc., we don't know yet. Our study seeks to better understand these processes.

2. "Patient-centered approaches have emerged in healthcare organizations seeking to move away from paternalistic approaches and encourage more active participation from patients, families, and communities as partners in the healthcare system." What are the theoretical frameworks or models that underpin the transition from paternalistic to patient-centered care in the context of QI processes, and how have these models been evaluated for effectiveness in previous studies?

A framework frequently published in the literature is Carman and colleagues' (2013) Multidimensional Framework For Patient And Family Engagement In Health And Health Care. Carman’s framework was influenced by the continuum of public participation in governance Arnstein’s “ladder of citizen participation” (Arnstein, 1969). Carman’s continuum of engagement conceptualizes patient engagement as an active partnership between patients, families or their representatives and healthcare professionals aimed at improving healthcare at three levels: individual care, organization and policy. Subsequently, the Université de Montréal drew inspiration from Carman's framework (2013) to create the Montreal Model (Pomey et al. 2015). Like Carman and colleagues (2013), the Montreal Model engages patients and their families in the individual care, organization and policy, but also in education and research.

A cluster randomized trial by Boivin and colleagues (2014) tested the impact of involving patients in setting healthcare improvement priorities for chronic care at the community level. Their conclusions report that patient involvement can change priorities driving healthcare improvement at the population level.

The systematic review by Bombard and colleagues (2018) investigated patient engagement in improving quality of care. This review identified several effective strategies and contextual factors that enable optimal engagement of patients in the design, delivery, and evaluation of health services. Their conclusions report that further data is needed to understand patients' experience of the engagement process.

3. "This qualitative multiple case study will be conducted in four large healthcare organizations in Quebec. Twelve to 15 key respondents will be recruited for each case." What criteria were used to select these healthcare organizations and respondents, and how do these criteria ensure a comprehensive understanding of the patient partnership approach across different organizational contexts?

According to Stake (2005), a case study should consist of 4 to 10 cases to benefit from the advantages of this design. 

This sentence has been added. See The cases in the Recruitment and sampling sub-section.

The Quebec healthcare network is divided into large care and service territories served by 22 integrated [university] health and social services centres (CI[U]SSSs). A CI[U]SSS is part of Quebec’s public healthcare network and is associated with one of its administrative regions. The size of the territory served varies from one CI[U]SSS to another. Those located in large territories must contend with a rural geographic context that can hinder accessibility to care and services. Some health and social services centres, specifically CIUSSSs, have a university affiliation. Along with this af

---

## [Decision Letter · Decision Letter 1]

2 Jul 2024

Implementing the Patient Partnership Approach to Quality Improvement of Care and Services: A Multiple Case Study Protocol

PONE-D-23-41748R1

Dear Dr. Deslauriers,

We’re pleased to inform you that your manuscript has been judged scientifically suitable for publication and will be formally accepted for publication once it meets all outstanding technical requirements.

Kind regards,

Yaodong Gu

Academic Editor

PLOS ONE

Additional Editor Comments (optional):

Well done!

Reviewers' comments:

Reviewer's Responses to Questions

**Comments to the Author**

1. Does the manuscript provide a valid rationale for the proposed study, with clearly identified and justified research questions?

Reviewer #1: Yes

Reviewer #2: Yes

2. Is the protocol technically sound and planned in a manner that will lead to a meaningful outcome and allow testing the stated hypotheses?

Reviewer #1: Yes

Reviewer #2: Yes

3. Is the methodology feasible and described in sufficient detail to allow the work to be replicable?

Reviewer #1: Yes

Reviewer #2: Yes

4. Have the authors described where all data underlying the findings will be made available when the study is complete?

Reviewer #1: Yes

Reviewer #2: Yes

5. Is the manuscript presented in an intelligible fashion and written in standard English?

Reviewer #1: Yes

Reviewer #2: Yes

6. Review Comments to the Author

You may also provide optional suggestions and comments to authors that they might find helpful in planning their study.

Reviewer #1: After careful revision by the authors, this manuscript has been greatly improved and all issues have been addressed.

Reviewer #2: Thank you to the authors for their hard work. The reviewers believe that after detailed revisions, this study has reached the level required for publication.

7. PLOS authors have the option to publish the peer review history of their article (what does this mean?). If published, this will include your full peer review and any attached files.

Reviewer #1: No

Reviewer #2: **Yes: **Zixiang Gao

---

## [Editor Report · Acceptance letter]

11 Jul 2024

PONE-D-23-41748R1 

PLOS ONE

Dear Dr. Deslauriers, 

I'm pleased to inform you that your manuscript has been deemed suitable for publication in PLOS ONE. Congratulations! Your manuscript is now being handed over to our production team.

Kind regards, 

on behalf of

Professor Yaodong Gu 

Academic Editor

PLOS ONE